# Dynamic Response and Geogrid Strain Analysis of GRS Retaining Wall

Jiaquan Wang [1,2], Wentao Zhong [1,2], Zhinan Lin [1,2,*] and Yi Tang [1,2]

1 College of Civil and Architectural Engineering, Guangxi University of Science and Technology, 2 Wenchang Road, Liuzhou 545006, China
2 Guangxi Zhuang Autonomous Region Engineering Research Center of Geotechnical Disaster and Ecological Control, 2 Wenchang Road, Liuzhou 545006, China
* Correspondence: zhinan_lin@gxust.edu.cn

**Abstract:** Modular Geogrid Reinforced Soil (GRS) retaining walls, as flexible structures, usually have a certain deformation capacity. However, the deformation damage of the facing panels will directly affect the durability performance of the retaining wall and pose a threat to the safety and operation of the road and related facilities. In order to study the influence of different load factors on the deformation mode and failure characteristics of the retaining wall, an indoor large-scale model test was carried out. The test load considers the average load, peak value, amplitude and frequency of load under traffic load. The changes in settlement and horizontal deformation, geogrid strain and acceleration response of the GRS retaining wall are compared and analyzed. The results show that in the dynamic test, the two wall damage modes are "wall facing outward tilt" and "wall facing outward curved". The maximum strain of the geogrid was 4.5% and 3.6%, respectively, which did not reach the damage strain. The peak load is the largest mechanical response of all load factors, followed by the load magnitude and average value, and finally the load frequency. In addition, combining the existing GRS retaining wall deformation and earth pressure calculation theory, a set of calculation methods for the strain of tendons under external load is proposed.

**Keywords:** GRS retaining wall; traffic load; model test; mechanical properties; geogrid strain

## 1. Introduction

GRS retaining wall is a common road slope retaining structure, which has more than 40 years of engineering use in China. Application of geosynthetic reinforcement technology can largely reduce the amount of steel and cement, reduce construction waste, reduce carbon emissions.

The basic structural form of a GRS retaining wall is shown in Figure 1. In a reinforced soil structure, the lateral pressure of the self-weight of the fill and the external load acts on the facing panels and passes through the panel to the geogrid, which is pulled outward, while the vertical earth pressure in the fill holds the geogrid down and increases the friction between the fill and the geogrid to prevent the geogrid from being pulled out. Therefore, without considering the deformation, the GRS retaining wall can remain stable as long as the geogrid has sufficient strength and generates sufficient frictional resistance with the soil. However, the GRS retaining wall is a flexible structure, which will produce certain deformation during service, according to field test investigations [1–7]. The main structural problems of GRS retaining walls include excessive lateral displacement, surface cracks and local collapse, and the causes of the problems are mainly attributed to insufficient fill compaction, excessive geogrid spacing, too short geogrid length and excessive dynamic loads. For this reason, stringent design methods [8–10] are specified in relevant guidelines and codes around the world to ensure the long-term internal, external and overall stability of GRS retaining walls. However, most design manuals tend to be conservative in determining geogrid design parameters, and the geogrid tension under normal operating loads is much

less than the design value for GRS retaining walls [11–15]. This indicates that there is still some room for improvement in the design of GRS retaining walls. Thus, Jacobs et al. [16] conducted tests on model retaining walls unreinforced and reinforced with geogrids of different tensile strengths, and found that backfills reinforced with geogrid had significantly lower earth pressures. The higher the reinforcement rate, the closer the location of the shear zone to the face, resulting in less earth pressure on the face. Scholars have carried out a lot of research on the mechanical properties of GRS retaining walls under static load, including indoor model tests [17–21], and numerical simulation studies [22–25]. Xiao et al. [26,27] conducted a series of model tests on GRS retaining walls to evaluate the effects of different influencing factors on the ultimate bearing capacity of GRS retaining walls under strip foundation, and initially explored the effects of top foundation location, geogrid length and panel connection method on the performance of retaining walls. Udomchai et al. [28] conducted full-scale tests on geosynthetic reinforced retaining walls to evaluate the working performance of the retaining wall during construction as well as in service condition, including settlement, bearing capacity, lateral displacement, earth pressure and geogrid stress. In addition, some scholars have also studied the constitutive model of GRS retaining walls under static load. Krystyna et al. [29] modified the existing specification method and proposed a prediction formula for the lateral displacement of the panel under static load, and analyzed the theoretical values compared with the experimental values through GRS retaining wall model tests, and illustrated the effect of reinforcement spacing and geogrid tensile strength on the calculation results. Desai et al. [30] predicted the vertical, horizontal stress and lateral displacement of GRS retaining walls under working stress by the Disturbed State Concept (DSC) finite element model, and compared with field tests, the finite element calculations were in good agreement with the field GRS mechanical behavior.

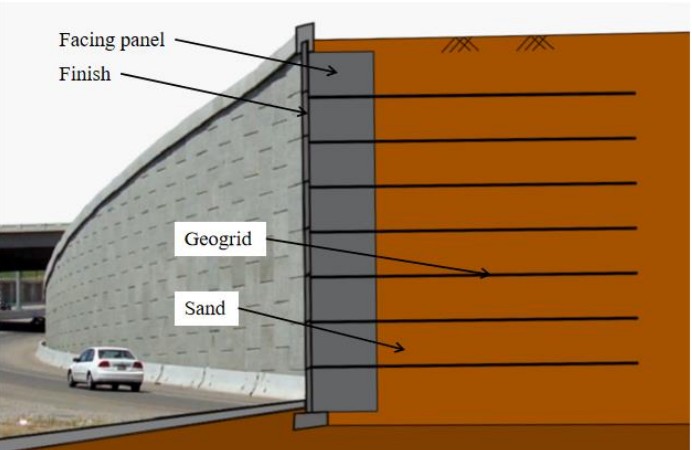

**Figure 1.** Structural form of geogrid reinforced retaining wall.

In practical engineering, traffic loads are more common and destructive than static loads [31]. To date, there are few research works on the working performance and dynamic response of GRS retaining walls. Wang et al. [32] investigated the effect of vehicle loading on the strain of geogrid and the effect of geogrid arrangement spacing and length on the performance of retaining walls. Ehrlich et al. [33] carried out a study on retaining walls with different compaction levels, and the results show that compaction leads to a significant increase in the horizontal stresses in the reinforced soil structure. Hore et al. [34] proposed a generalized equation for the prediction of normalized horizontal displacements by compaction under dynamic loads. Liu et al. [35] studied the additional stresses in the soil caused by vehicle loads, and found that the additional stresses were distributed nonlinearly along the wall height, with the peak located in the middle of the retaining wall. It can be seen that the research on the deformation and damage mechanism of the GRS retaining wall under static load is relatively mature, while the research on its deformation under traffic load and related theories is still in the initial stage.

Considering that it is too costly to carry out the damage testing of retaining walls on site and limitations due to site conditions, it is extremely difficult to carry out in situ tests. In contrast, the indoor model test can more directly respond to the change law of mechanical response of the retaining wall. In this paper, the deformation and damage characteristics of GRS retaining walls under different loading methods are compared and analyzed by indoor model tests, respectively. The characteristics of the influence of different loading factors on the test law of GRS retaining walls are studied, and the calculation method of geogrid strain under external loading is investigated.

## 2. Model Tests

### 2.1. Test Model

The structural form of the test model is a modular geogrid reinforced retaining wall. In order to make the model test results truly reflect the mechanical behavior of real GRS retaining walls under traffic loads, the similarity relationship of the test model was determined according to the dimensions of the test setup. The relationship between the variables through the method of dimensional analysis is derived with geometric similarity ratio $C_1 = 4$, filler volumetric weight similarity ratio $C_\gamma = 1$, cohesion similarity ratio $C_C = 1$, friction angle similarity ratio $C_\varphi = 1$, geogrid tensile modulus similarity ratio $C_E = 2$.

The test model box is a homemade 3.0 m × 1.6 m × 2.0 m ($L \times W \times H$) large model box, as shown in Figure 2. The model box frame is made of 0.6 cm thick channel steel welded together and welded with channel steel of equal thickness in both longitudinal and transverse directions to prevent deformation of the box sides during loading. The model box is installed with double-layer 1cm thick tempered glass on one side to facilitate test observation, and a retaining wall facing panel is installed on the front, with a retaining wall height of $H$ = 1.8 m.

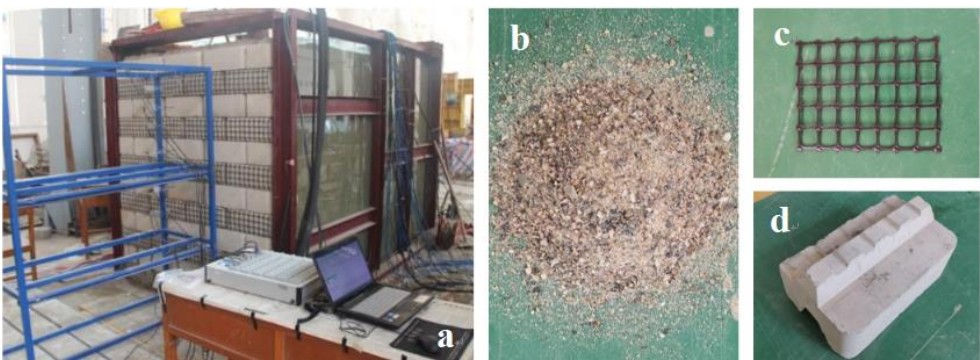

**Figure 2.** Preparation: (**a**) test site layout; (**b**) sand; (**c**) geogrid; (**d**) panel.

Referring to the specification (Technical Specifications for Construction of Highway Subgrades [36]), every 15 cm of fill was compacted, the quality and height of each layer of fill were strictly controlled and the compaction of fill was not less than 95%. When filling, 20 kg weights were first used to manually compact 3 times, then an electric plate compactor was used to tamp and level. In order to prevent the position and direction of the test element from changing due to tamping during the filling process, the soil layer in which the measuring element was buried was tamped first, and then the measuring element was buried at each monitoring point. After backfilling the soil in the recess, the area was compacted twice. In order to make the geogrid connect firmly with the panel, the wall was installed with a reverse wrapped panel, reinforcement spacing $s$ = 0.3 m (2 panel heights) and the buried length of each layer of geogrid was $L$ = 2.0 m. The test filling process is shown in Figure 3, the geogrid and monitoring instrument arrangement is shown in Figure 4.

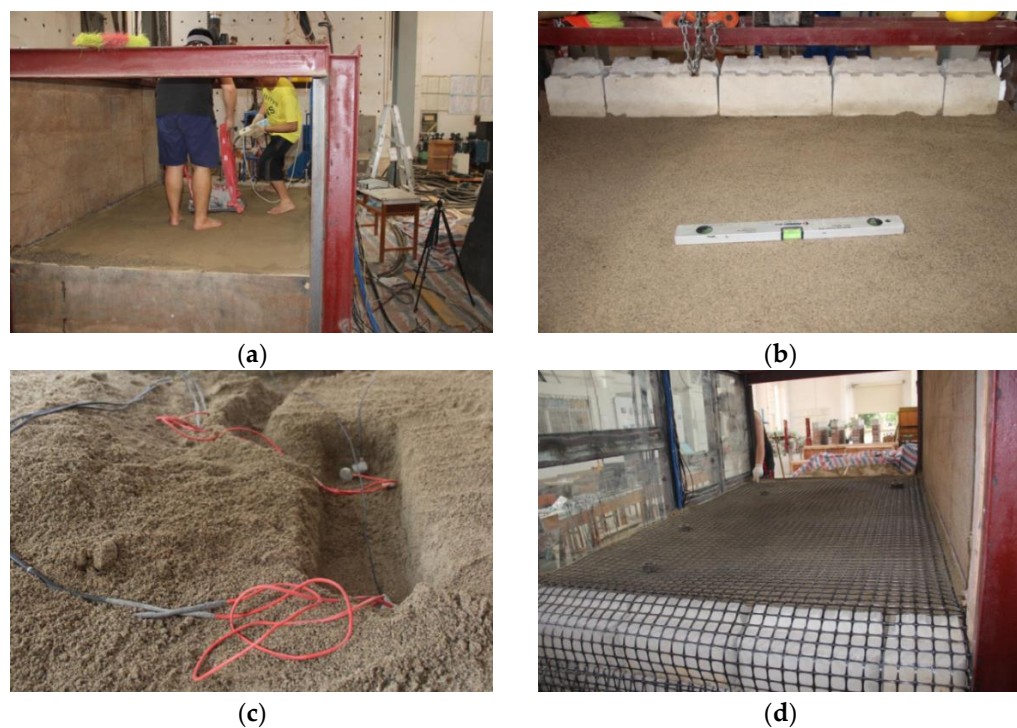

**Figure 3.** Filling process: (**a**) tamped backfill; (**b**) flattening; (**c**) burying measuring elements; (**d**) laying geogrid.

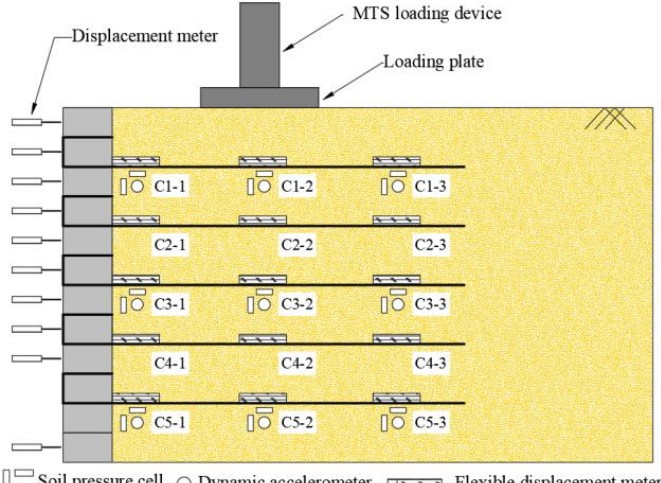

**Figure 4.** Instrument arrangement diagram.

*2.2. Test Materials*

The test soil was obtained from local river sand in Liuzhou, Guangxi. The particle gradation curves of the soil samples measured by indoor sieving tests are shown in Figure 5. The inhomogeneity coefficient was 8.44, curvature coefficient was 1.15 and, according to the Unified Soil Classification System [37], the test soils were classified as well-graded sand (SW). At the time of the test, the moisture content of the soil sample under natural conditions was 4.6%, the angle of internal friction was 35° and the maximum dry density was 1.81 g/cm$^3$. Facing panels were made of precast concrete blocks (concrete grade was C35). The geogrid used in the test was a two-way stretch plastic geogrid, and the performance of the geogrid was determined by standard tensile tests [38], as shown in Table 1.

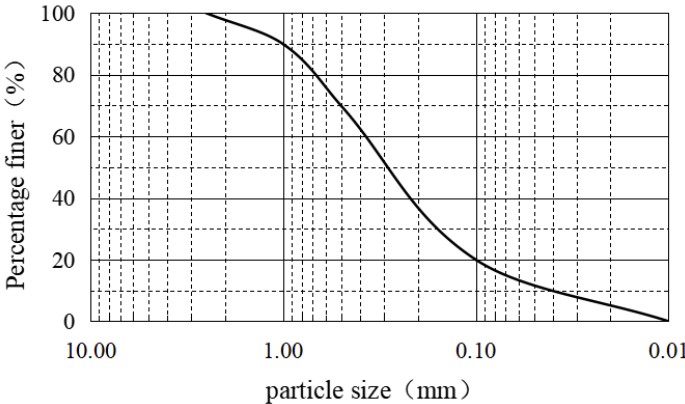

**Figure 5.** Soil grading curve.

**Table 1.** Specific parameters of geogrid.

| Item | Value |
|---|---|
| Longitudinal ultimate tensile strength (kN/m) | 31.4 |
| Transverse ultimate tensile strength (kN/m) | 32.5 |
| Longitudinal yield elongation (%) | 13.4 |
| Transverse yield elongation (%) | 13.6 |
| Tensile strength at 2% elongation in the longitudinal direction (kN/m) | 13.7 |
| Tensile strength at 2% elongation in the transverse direction (kN/m) | 14.0 |
| Tensile strength at 5% elongation in the longitudinal direction (kN/m) | 24.3 |
| Tensile strength at 5% elongation in the transverse direction (kN/m) | 24.7 |
| Aperture size (mm × mm) | 36 × 40 |

### 2.3. T*est Scheme*

The loading equipment used in the test was the electro-hydraulic servo loading system of MTS Company in the United States, which can respond to the output pressure load and loading frequency in real time, and can achieve static or dynamic application of load by regulating the relevant parameters. In the dynamic test, the vertical dynamic stress generated by the moving vehicle in the road bed is used as the main test object, and the half-sine wave is applied through the loading plate (60 cm × 20 cm) to simulate the traffic load generated by the moving vehicle [39–41]. The value of loading value $P$ for half-sinusoidal function is as follows:

$$P = P_0 + P_A sin(2\pi ft), \tag{1}$$

where $P_0$ is the average load (kPa); $P_A$ is the load amplitude (kPa); $f$ is the frequency of load (Hz).

The loading method of this test load considered the effect of peak value, average value, amplitude value and frequency of dynamic load on the deformation damage of the retaining wall, and further analyzed the sensitivity of different loading factors on the mechanical response of the GRS retaining wall. A series of orthogonal tests with different peak, average, amplitude and frequency of dynamic load, resulting in two different loadings, were conducted. One is the same frequency loading method, keeping the loading frequency of 2 Hz unchanged, the dynamic load of each stage lasts for 10 min and the loading plate is gradually loaded until the inclined instability occurs, which is regarded as the failure of the GRS retaining wall. Another one is the same amplitude loading method, starting from $10 \pm 10$ kN, with each load corresponding to four different frequencies of 2 Hz, 4 Hz, 6 Hz and 8 Hz. Under each load frequency, the loading lasts for 10 min, and the average value of the next load increases by 20 kN, thus forming a gradual growth relationship of 0~20 kN, 20~40 kN, 40~60 kN, etc. until destruction. The specific output mode of dynamic load is shown in Figure 6.

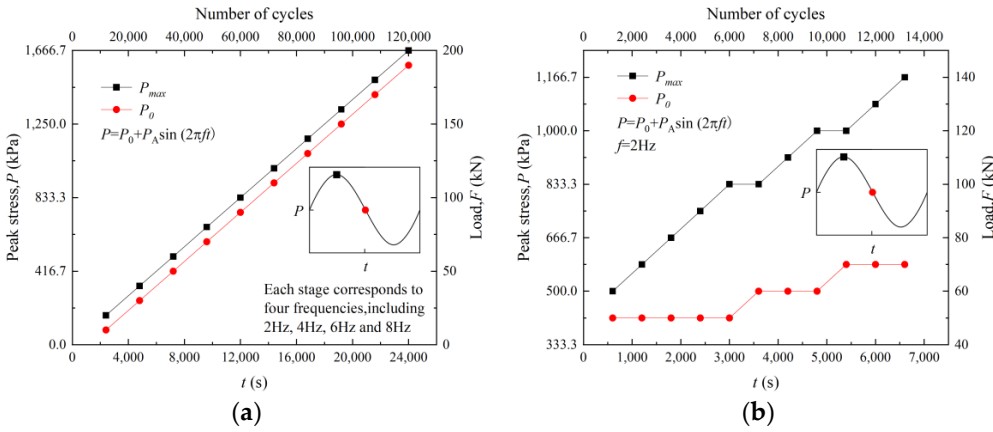

**Figure 6.** Dynamic load loading method. (**a**) Equal amplitude loading. (**b**) Equal frequency loading.

## 3. Test Results and Analysis

### 3.1. Lateral Deformation of Wall Facing

Figure 7 shows the variation in lateral displacement of the GRS retaining wall panel under dynamic load versus wall height $H$, where the equal amplitude loading retains only the lateral displacement at 8 Hz except for the dynamic load of $190 \pm 10$ kN. From the figure, it can be seen that the lateral displacement of the retaining wall increases with the increase in the peak load and finally reaches the limit state. Numerous test results have been obtained worldwide showing that the retaining wall deforms more under dynamic load compared to static load. This is due to the fact that the impact of the loading plate on the soil below is more violent under dynamic load, the repeated tension and relaxation of the uppermost geogrid make the embedded locking effect between the geogrid and the soil particles not strong under static load, which will affect the restraint effect of the geogrid on the horizontal direction of the upper soil.

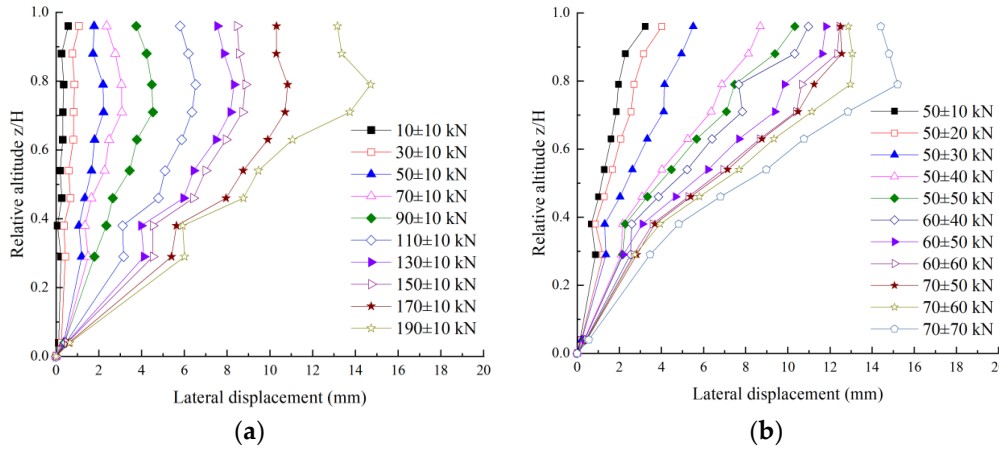

**Figure 7.** Lateral deformation of wall facing. (**a**) Equal amplitude loading. (**b**) Equal frequency loading.

During the loading process, the two groups of test geogrids were arranged in the same way, but the damage patterns were obviously different. In the case of constant amplitude, the lateral displacement of the facing panels in Figure 7a was "wall facing outward curved", and the deformation range was between $0.4H$ and $0.95H$ of the retaining wall. The authors concluded that the lower amplitude dynamic load makes the reinforced soil structure denser through vibration and strengthens the soil integrity, so that the GRS retaining wall can withstand higher peak dynamic loads, and with the increase in the peak dynamic load, the lateral additional stress in the middle of the retaining wall gradually increases [37], and finally leads to bulging damage of the retaining wall. As shown in Figure 7b, the lateral displacement of the facing panel is "wall facing outward tilt" at constant frequency,

and the lateral displacement is maximum at 0.83*H*. The reason is that during the loading process, the increase in dynamic load amplitude makes the impact of the loading plate on the reinforced soil structure intense, and the uppermost layer of the geogrid is repeatedly tensioned and relaxed, thus the embedded locking effect between the geogrid and soil particles is weakened. With the increase in peak dynamic load, the loading plate gradually sinks, so that the soil below the loading plate spreads to both sides, and the extruded soil imposes additional lateral additional stress on the upper panels. This is the main reason for the increasing lateral displacement of the upper panels of the retaining wall, which is expressed as "wall facing outward tilt".

In addition, from the results of this test, among the two wall damage modes, the ultimate bearing capacity of "wall facing outward curved" is $1.5 \times 10^3$ kPa ($F = 170 \pm 10$ kN) higher than that of "wall facing outward tilt" of $1.08 \times 10^3$ kPa ($F = 70 \pm 60$ kN). The authors believe that the role of the panel is mainly to ensure the local stability of the fill near the panel, the deformation mode of "wall facing outward curved" is multiple panels to bear the deformation, which can improve the bearing capacity, while the stress of "wall facing outward tilt" is concentrated in the upper part of the retaining wall, leading to the reduction in bearing capacity.

Figure 8 shows the relationship between the lateral displacement of the facing panels and the magnitude and frequency of the dynamic load. From the figure, it can be seen that the lateral displacement is influenced more by the peak dynamic load and less by the frequency and the number of load cycles, and continuing to increase the average dynamic load will increase the effect of frequency on the lateral displacement. Below the height of 67.5 cm, the lateral displacement is significantly smaller than that of the upper part, indicating that the dynamic load has little effect on the soil below 1.2 m from the top of the wall.

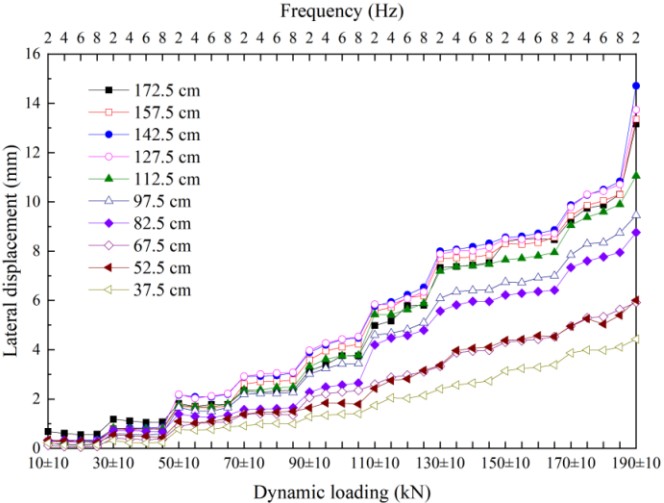

**Figure 8.** Variation in lateral displacement and frequency.

### 3.2. Geogrid Strain

The above sections mainly shows that the good engineering properties of GRS retaining walls depend mainly on the interaction between tendons and soils. However, under long-term loading, geogrids can deform and affect the working performance of GRS retaining walls [42–45]. In order to study the mechanical response law of geogrids under the action of traffic load, three flexible displacement meters at different horizontal distances from the retaining wall panel were selected to record the geogrid strain, and the results are shown in Figures 9 and 10. It can be seen from the figure that along the height direction of the retaining wall from bottom to top, the distribution of geogrid strain shows a growing trend, and the peak dynamic load has a large effect on the geogrid strain, which is consistent with the trend of lateral displacement of the retaining wall. As the reinforcement depth increases,

the strain of the last layer of the geogrid is relatively small and does not give full play to the performance of the geogrid. This is consistent with the test law of Keskin et al. [46].

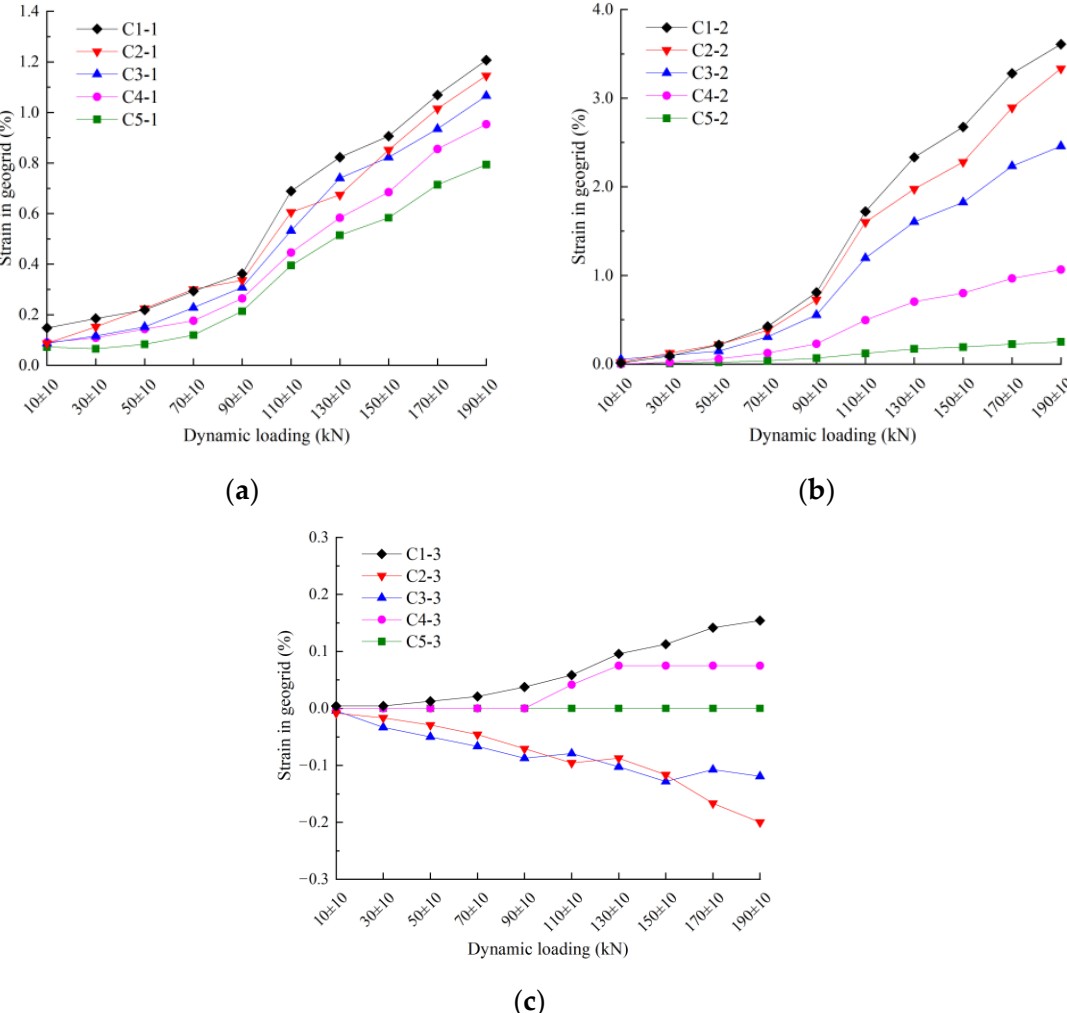

**Figure 9.** Geogrid strain (equal amplitude loading). (**a**) Connection position to the facing panels. (**b**) Below the loading plate. (**c**) Anchorage position.

From Figures 9 and 10, at the connection position to the facing panels (C1-1 to C5-1), where the strain of the geogrid is mainly controlled by the horizontal displacement of the facing panels, the strain of the geogrid at the maximum lateral displacement is also maximum, but the strain is small. In addition, the soil near the facing is not easily compacted under the action of dynamic load, so the embedded ability of the geogrid to the surrounding soil is weak. Once there is a gap between the facing and the surrounding soil, the geogrid will bear the shear stress and stress concentration, which is caused by the uneven settlement of the soil, resulting in a large strain. This is extremely evident when the stress peaks reach $1 \times 10^3$ kPa ($P = 110 \pm 10$ kN) (Figure 9) and 750 kPa ($P = 50 \pm 40$ kN) (Figure 10). This mechanical behavior at the location of the connection between the panel and the geosynthetic material is in general agreement with the "stress concentration–separation–deformation" phenomenon of Lu et al. [47].

Below the loading plate (C1-2 to C5-2), the strain of the geogrid increases with the increase in the peak load. As can be seen from Figure 10, when keeping the peak load constant and changing the average value and amplitude of the load, almost no strain occurs in the geogrid in the soil, indicating that the strain in the geogrid is mainly affected by the peak load. In addition, the mechanical response of the geogrid under both loading methods was maximum at C1-2, and the maximum value of geogrid strain reached 4.5% and 3.6%,

indicating that the actual tensile stress on the geogrid did not reach its tensile strength. Yang et al. [48], through a field test of a geogrid GRS retaining wall for a passenger line, measured that the tensile force of the geogrid during service is much less than its tensile strength value, and the maximum strain accounts for less than 30% of the peak strain. However, despite this, the strain of the uppermost geogrid still accounts for more than 30% of the total strain, which means that the geogrid nearest to the loading plate takes on the main job of controlling settlement and plays an important role in the foundation bearing capacity and overall stability of the retaining wall.

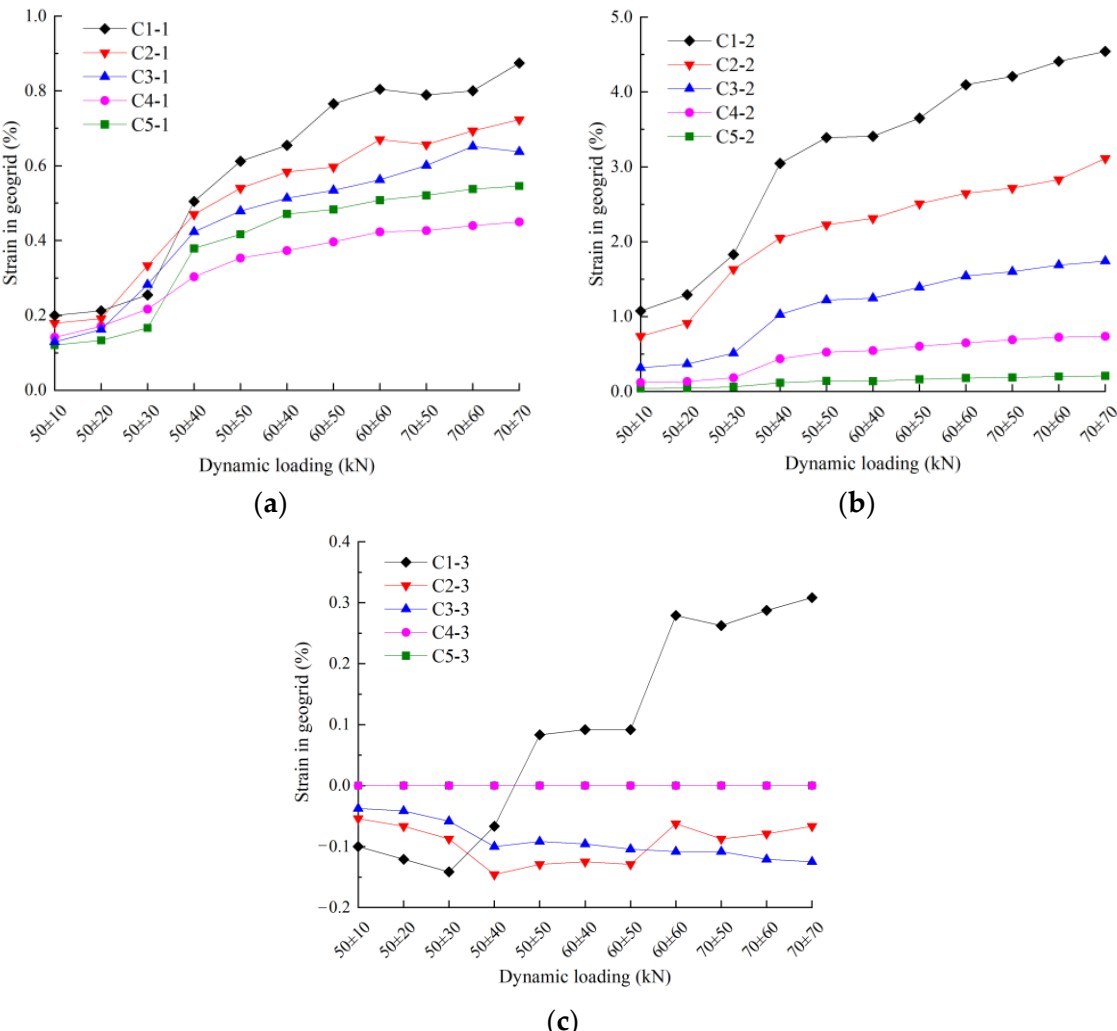

**Figure 10.** Geogrid strain (equal frequency loading). (**a**) Connection position to the facing panels. (**b**) Below the loading plate. (**c**) Anchorage position.

At the anchorage position in the geogrid (C1-3 to C5-3), the strains in all layers are not obvious except for the upper geogrid, where the strain tends to increase. The closer to the end of the geogrid, the smaller the reinforcement effect, so that the reinforced area and non-reinforced area forms an excess, not producing a large uneven settlement. In addition, the strain of the geogrid at C3-3 and C2-3 showed negative growth. The reason is that the geogrid is strained by the self-weight of the soil and the pressure of mechanical compaction during the filling process, and this strain is small and is an elastic strain. During the loading process, the soil particles rearrange and the geogrid has a certain space to rebound and shrink, resulting in a negative value as the flexible displacement meter reading decreases.

### 3.3. Acceleration Response

As shown in Figures 11 and 12, the acceleration response of soil under dynamic load was measured by a dynamic accelerometer in the test, and the peak value of dynamic response was taken to study the influence of average load, peak value, amplitude and frequency on dynamic characteristics of soil under dynamic load. The preliminary study found that the peak acceleration in the soil decreases from top to bottom. Reinforced soil structure has a better acceleration weakening effect on high-frequency loads than on low-frequency loads. This is because the soil under dynamic load is equivalent to a damper, and the frictional collision of soil particles causes the energy to be consumed as a way to reduce the acceleration response in the soil. The farther away from the vibration source, the more obvious the damping effect was.

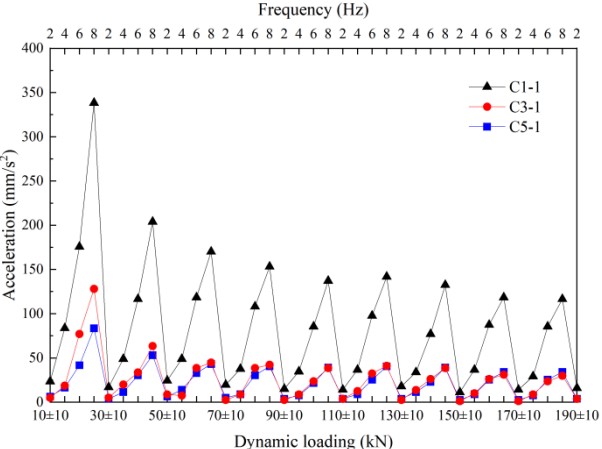

**Figure 11.** Variation in acceleration peak and frequency.

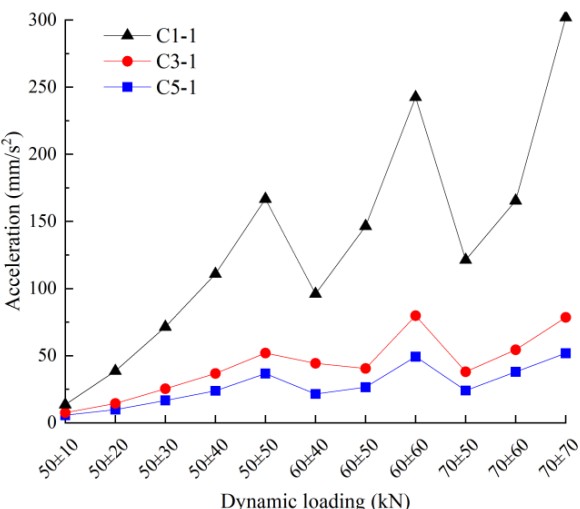

**Figure 12.** Distribution pattern of acceleration peaks at different depths.

It can be found from Figures 11 and 12 that increasing the load amplitude and frequency both increase the peak acceleration. In the case of constant amplitude loading (Figure 11), by keeping the peak load unchanged and increasing the frequency, the acceleration peak will increase, while by keeping the load frequency unchanged and increasing the peak, the peak acceleration changes little. In the case of constant frequency loading (Figure 12), by keeping the average value unchanged and increasing the amplitude, the acceleration peak will increase, while by keeping the amplitude unchanged and increasing the average value, the acceleration peak will decrease. This shows that the different load factors influencing the peak acceleration of the GRS retaining wall under traffic load are the load frequency, amplitude, peak and average value. The reason for this is that the

acceleration response is different from the dynamic response, and the peak acceleration does not affect the deformation of the retaining wall. The greater the acceleration, the faster the energy consumption. Feng et al. [49] proposed in their study that the geogrid will produce a "stretching effect" and the "dynamic tension membrane effect" under traffic load, and this mechanism will increase the action area of the geogrid and soil particles, and the greater the frequency and magnitude of the traffic load, the more obvious the effect.

It thus appears that the acceleration response in the soil is a mechanism for coordinating deformation in a reinforced soil structure. Under dynamic load, the soil particles move irregularly. During the movement, the direct collision friction of soil particles and the friction between the geogrid and soil particles transfer and dissipate the energy in the soil. In this process, the compactness of the reinforced soil structure is improved. Taking equal frequency loading as an example (Figure 12), with the increase in the number of load cycles, C3-1 and C5-1 show an increasing decay of the acceleration peak compared to C1-1. This indicates that as the number of load cycles increases, the compactness of the upper reinforced soil structure gradually becomes higher than that of the surrounding soil, and thus the range of acceleration propagation in the soil becomes smaller.

## 4. Geogrid Strain Calculation Method

### 4.1. Basic Assumptions

The analysis method of geogrid tension established in this paper is based on the following assumptions:

(1)  geogrids are laid horizontally and remain horizontal under the action of soil self-weight;
(2)  the additional stress acting on the pressurized zone of the geogrid distributes uniformly along the horizontal direction;
(3)  the tension caused by compaction of the geogrid during filling is ignored;
(4)  under the action of self-weight and external load, the deformation of the geogrid is mainly elastic deformation, and the plastic deformation is small and can be ignored.

Assumption (1) above is consistent with the reality for most reinforced earth retaining walls. A large number of large-scale tests and field monitoring results in China and abroad show that the geogrid strain is not large under normal working load. Under the condition of guaranteed construction quality, the geogrid strain caused by the self-weight of filling soil can be neglected. Assumption (2) is based on assumption (1), that the vertical additional stress generated by the external load in the soil decreases gradually with the increase in depth. The calculation of the vertical additional stress of layer $i$ of the geogrid can be carried out by the simplified method of 45° diffusion angle (as shown in Figure 13). The larger the $z_i$ is, the larger the diffusion length $l_i$ of the vertical additional stress is, and it is evenly distributed along the tensile direction of the geogrid. Regarding assumption (3), it is mentioned above that the geogrid will be deformed during filling and compacting, and this deformation is very small and can recover by itself some time after the filling is completed, which has no effect on the calculation results. From the strain of each layer of the geogrid in the test, the maximum value of strain does not exceed 5%, and the tensile strength of the geogrid is not fully exerted. It is reasonable to assume that the deformation of the geogrid conforms to Hooke's law (4) [50].

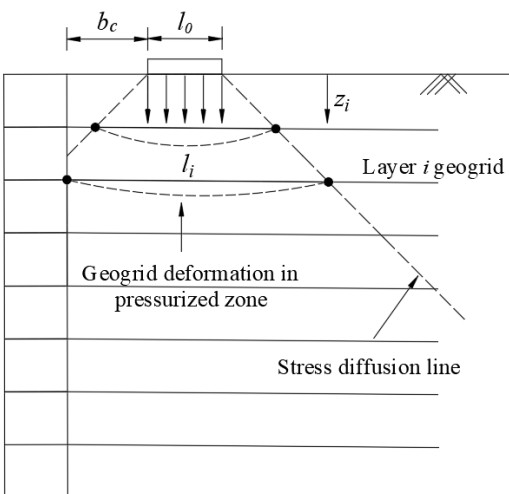

**Figure 13.** Deformation diagram of geogrid in pressurized zone.

### 4.2. Derivation of Calculation Formula

As shown in Figure 14, for the analysis of the compression zone of the layer *i* geogrid, establishing a Cartesian coordinate system, the displacement equation $y(x)$ is established based on the average relationship of the forces after deformation of the AB section. The component force $T_x$, $T_y$ of the geogrid tension $T$ in the x-axis and y-axis directions is

$$\{T_x(x),\ T_y(x)\} = \left\{F_H,\ F_H \frac{dy}{dx}\right\}, \tag{2}$$

where $F_H$ is the design value of horizontal tension of the geogrid.

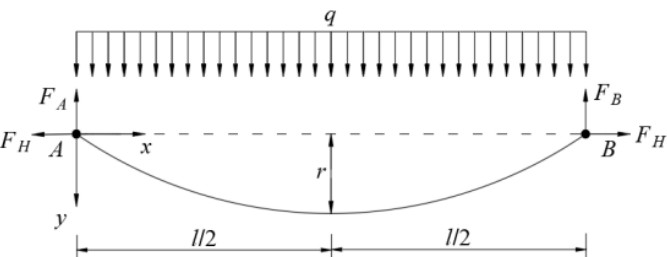

**Figure 14.** Geogrid deformation in pressurized zone.

The force analysis of one unit of the geogrid in the pressurized zone is shown in Figure 15.

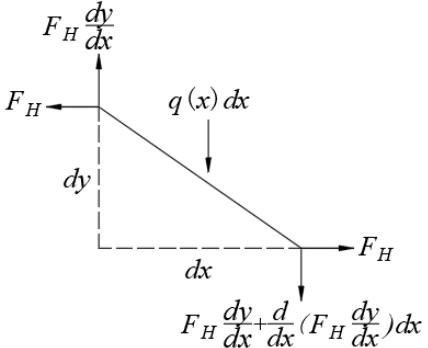

**Figure 15.** Force analysis of unit.

According to the balance of force, the differential equation is obtained:

$$\sum F_y = 0, \quad F_H \frac{d^2 y}{dx^2} + q(x) = 0. \tag{3}$$

Two integrals are available:

$$y = \frac{q}{2F_H} \cdot x \cdot (l - x) + \frac{c}{l} x. \tag{4}$$

Using in the boundary conditions available, $c = 0$, the displacement equation of the AB section is

$$y(x) = \frac{q}{2F_H} \cdot x \cdot (l - x). \tag{5}$$

Bringing the point $(x, y) = (l/2, r)$ into Equation (5), the expression of mid-span deflection $r$ is

$$r = \frac{ql}{8F_H}. \tag{6}$$

By the Pythagorean theorem, the expression for the variation in the geogrid tension $T$ with $x$ can be found

$$T = F_H \sqrt{1 + \left(\frac{dy}{dx}\right)^2} = F_H \sqrt{1 + \left(\frac{8f}{l^2}\left(\frac{l}{2} - x\right)\right)^2}. \tag{7}$$

For a general GRS retaining wall, the vertical stress on layer $i$ of the geogrid is

$$q = \gamma z_i + \frac{F}{l_i}, \tag{8}$$

where $\gamma$ is the volumetric weight of the filler, $z_i$ is the distance from the top of the wall to layer $i$ of the geogrid, $F$ is the peak load, and $l_i$ is the length of the pressure zone of layer $i$ of the geogrid, calculated according to the following formula.

When $z_i \leq b_c$,

$$l_i = l_0 + 2z_i. \tag{9}$$

When $z_i > b_c$,

$$l_i = l_0 + b_c + z_i, \tag{10}$$

where $l_0$ is loading plate width, $b_c$ is distance between loading plate and facing.

Tensile forces at the nodes of the geogrid are calculated with reference to the current relevant codes [51]:

$$F_H = K p_i s_x s_y. \tag{11}$$

In the formula, $K$ is the additional coefficient of the peak tension of the geogrid, 1.5~2.0, $p_i$ is the horizontal earth pressure, $s_x$, $s_y$ is the horizontal and vertical spacing between the geogrids.

Bringing $F_H$, $q$ into Equation (5), and according to the geometric relationship, the elongation of the geogrid can be obtained:

$$\Delta l = \int_0^l \sqrt{1 + \left(\frac{dy}{dx}\right)^2} \, dx - l. \tag{12}$$

Therefore, the geogrid strain $\varepsilon$ is

$$\varepsilon = \frac{\Delta l}{l}. \tag{13}$$

A large number of large-scale tests and field monitoring results in the world show that increasing the tensile strength of the geogrid can reduce the settlement under external load.

However, in the calculation method established in this paper, the deformation deflection *r* of the geogrid does not consider the elastic modulus of the geogrid, and the result thus obtained is obviously inconsistent with the actual situation.

According to Equation (6), if we let r decrease, the tension $F_H$ at the geogrid node should increase. Thus, it is found that the derivation of the specification for $F_H$ is imperfect, $F_H$ is the internal tension of the geogrid and the specification only considers the horizontal earth pressure there, and the correct derivation is as follows.

The modulus of elasticity *E* of the geogrid is calculated by a tensile test of geosynthetic material.

$$E = \frac{\Delta F}{\Delta L} = \frac{F_d/s}{(L_c - L_a)/h},$$　　(14)

where: *E* is the modulus of elasticity, $\Delta F$ is the force per unit area on the cross section, $\Delta L$ is the ratio of elongation to clamping length *h*. The geogrid deformation can be found according to Hooke's law that

$$\Delta l = \frac{1}{l} \int_0^l \frac{T}{Et_0} dx,$$　　(15)

where: *E* is the linear modulus of elasticity, *t* is the thickness of the geogrid, this paper takes $t_0 = 0.2$ cm.

Combining Equations (13) and (15),

$$F_H = Elt \left( 1 - \frac{l}{\int_0^l \sqrt{1 + \left(\frac{dy}{dx}\right)^2} dx} \right).$$　　(16)

So far, all the unknown quantities of the equation have been solved. Bringing $F_H$ into *y(x)*, through Equations (7), (15) and (16), The geogrid strain $\varepsilon$ is obtained.

### 4.3. Test Verification

In order to verify the rationality of the strain calculation method of the geogrid under external load proposed in this paper, a set of model tests under static load conditions were conducted. After the filling was completed, the loading was started from 10 kN step by step until *F* = 140 kN, and the load was increased by 10 kN at each level and continued for 15 min, until the retaining wall was damaged. The lateral displacement of static load, with equal amplitude loading and equal frequency loading *F* = 140 kN, is shown in Figure 16.

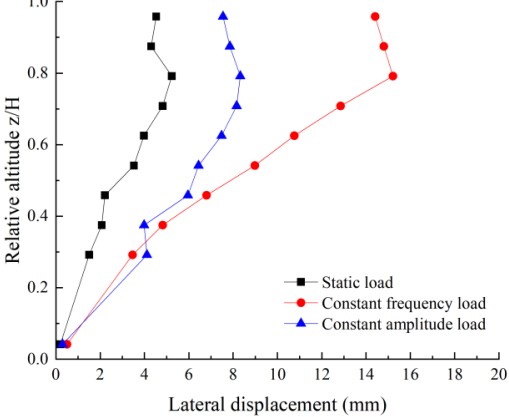

**Figure 16.** Lateral displacement at *F* = 140 kN.

Two groups of tests with static and equal amplitude loading were selected among them, the strains of the geogrid under each level of loading were calculated, the calculated values of geogrid strain were compared with the measured values and the results are shown in Figure 17. From the figure, it can be seen that the calculated and tested values have

good agreement, and it is feasible to calculate the geogrid strain under external loading by applying the calculation method in this paper.

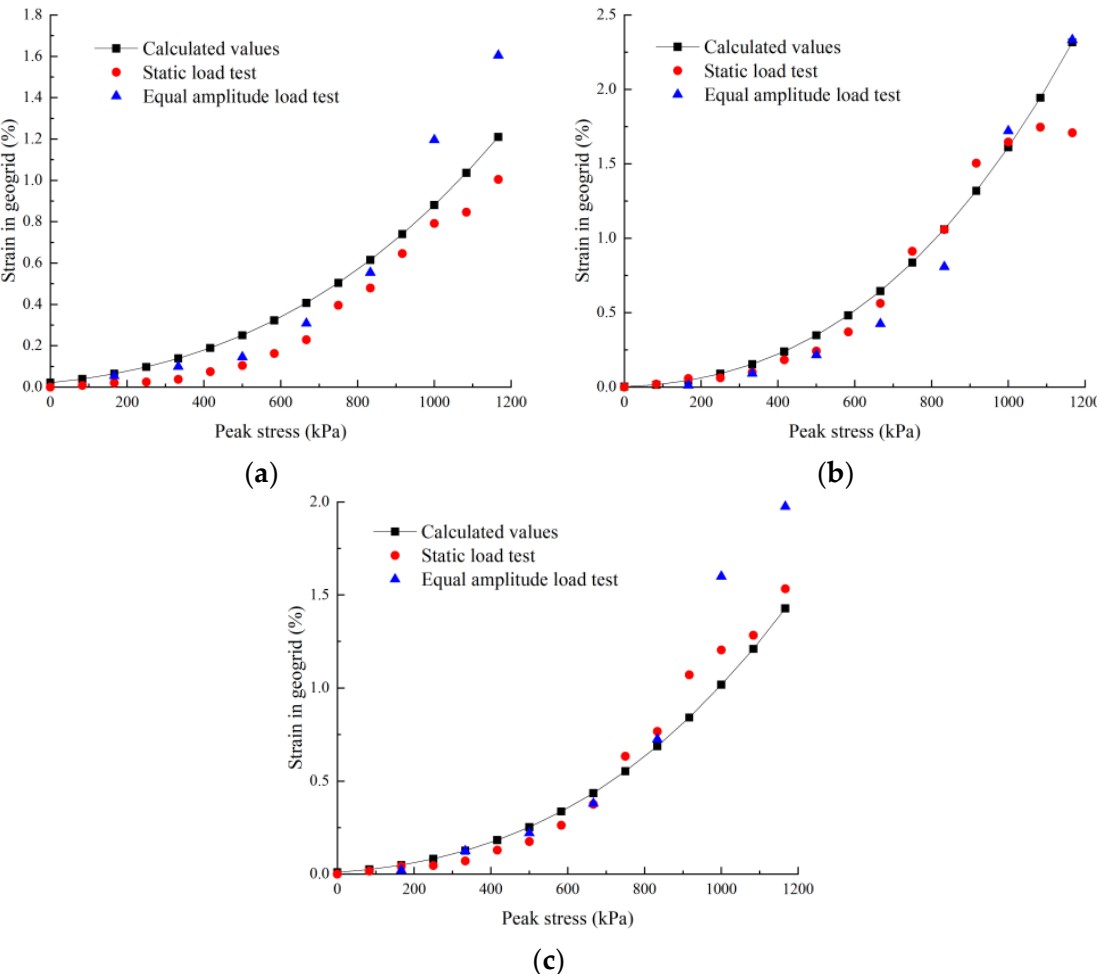

**Figure 17.** Comparison of calculated and test values. (**a**) Layer 1 geogrid. (**b**) Layer 2 geogrid. (**c**) Layer 3 geogrid.

This method can be used for the calculation of geogrid strain in general reinforced soil structures, which provides a new method for calculating the design value of geogrid tension in the relevant codes, and has certain guiding significance for improving the design calculation method of GRS retaining walls. However, there is room for improvement. The method proposed in this paper is only reliable under low-amplitude dynamic load and static load. However, for high-amplitude dynamic load, the embedded locking effect between the geogrid and soil particles is weakened, which will lead to larger displacement of the facing panels and deformation of the geogrid in the soil by pulling. How to consider these factors in the calculation analysis needs to be further studied.

## 5. Conclusions

This paper is based on a large-scale model test to investigate the dynamic characteristics of GRS retaining walls. The results of the study contribute to revealing the effects of peak, average, amplitude and frequency of loads on the mechanical response and damage characteristics of GRS retaining walls under dynamic loading, and provide guidance for the design of GRS retaining walls. The main conclusions are summarized as follows:

(1) In dynamic loading, the peak load is the most significant in the mechanical response of all load factors, followed by the amplitude and average value and finally the frequency. The peak load directly affects the additional stress in the soil, which

deforms the retaining wall facing panels and the geogrid; the load magnitude and average value both change the deformation characteristics of the wall by changing the effect between the geogrid and the soil particles, and then change the deformation characteristics of the wall facing; the load frequency changes the acceleration in the soil, but the soil particles consume it through frictional collision, so the degree of action is very small.

(2)   The dynamic load of lower amplitude makes the reinforced soil structure more compact through vibration, and the additional stress in the soil is mainly concentrated in the middle of the retaining wall, the bearing capacity of GRS retaining wall is increased and the deformation trend of the wall facing is "wall facing outward curved"; high-amplitude dynamic loads weaken the embedded locking effect between the geogrid and the soil, and the deformation of the wall surface is mainly concentrated in the middle and upper part, which shows "wall facing outward tilt", and the bearing capacity of GRS retaining wall is reduced.

(3)   The strain of the geogrid is mainly affected by the peak load, and its strain gradually becomes larger along the depth direction, and increases and then decreases along the horizontal direction. The geogrid nearest to the loading plate takes the main role in controlling the settlement, and plays an important role in the foundation bearing capacity and overall stability of the retaining wall.

(4)   Based on the basic theory of elastic mechanics and the deformation principle of geogrid reinforced earth retaining walls, a set of geogrid strain calculation methods under external load is established, and it provides a new method for calculating the design value of geogrid tension in the design calculation of GRS retaining walls.

**Author Contributions:** Paper revision and polishing, J.W. and Z.L.; Laboratory experiment, W.Z.; Paper writing, W.Z.; Literature research, Y.T. All authors have read and agreed to the published version of the manuscript.

**Funding:** The project was funded by the National Natural Science Foundation of China (No. 41962017), the Natural Science Foundation in Guangxi Province of China (No. 2022GXNSFDA035081, 2021GXNSFBA196043), the High-Level Innovation Team and Outstanding Scholars Program of Guangxi Institutions of Higher Learning of China (GuiJiaoRenCai[2020]6), the Doctoral Foundation of Guangxi University of Science and Technology (No. 03200009) and the Project to Enhance the Basic Research Ability of Young and Middle-aged Teachers in Guangxi Universities (No. 2020KY08023).

**Institutional Review Board Statement:** Not applicable.

**Informed Consent Statement:** Not applicable.

**Data Availability Statement:** The data presented in this study are available on request from the corresponding author.

**Conflicts of Interest:** The authors declare no conflict of interest.

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
