# Peer review of "Dynamic Response and Geogrid Strain Analysis of GRS Retaining Wall"

_applsci, doi:10.3390/app12199930_

Round 1

Reviewer 1 Report

This manuscript focuses on influence of different loading factors on the mechanical properties of GRS retaining wall, and discusses the influence of average, peak, amplitude and frequency of traffic load on the settlement, horizontal deformation, geogrid strain and acceleration response of GRS retaining wall, and proposes a set of calculation methods for the strain of geogrid under the action of external load. The findings are compelling and beneficial for the practitioners in this field. I found the paper was well-structured and informative. I gladly recommend it for publications in the applied sciences journal. However, there are still some minor concerns:

1.      The English of the paper should be improved. There are some places where grammar should be corrected. Please carefully check and correct throughout the manuscript.

2.      The abstract lacks the quantitative presentation of main results. The authors are required to add more specific information regarding the effect of various parameters to this part.

3.      The title of the paper is: Dynamic response and geogrid strain analysis of GRS retaining wall, a large part of the paper to study the calculation method of geogrid strain, but there is no such part in the key words, it is suggested to add.

4.      Page 3: The headings of the subsections in section 2 are not clear enough, for example, the test material should be placed in a separate subsection to make it easier for the reader to read.

5.      In subsection 2.1, only the grading parameters are given for the retaining wall filler, and the basic mechanical property index of sand filler is missing, and whether the sand is dried or the specific moisture content is not clearly stated when filling, please add.

6.      In subsection 2.2, the dynamic load loading method (Figure 5), the meaning of Pmax is not explained, please add, in addition, it is recommended to add the waveform of half sine wave.

7.      In subsection 3.2, it is mentioned that the horizontal distance from the panel at 15cm, 75cm and 135cm cannot be found in the text with its corresponding location, so it is recommended to use the sensor's mark directly.

8.      In Subsection 4.2, how does Figure 15 satisfy the equilibrium relation for the y-axis?

Author Response

Dear reviewers,

Thank you very much for taking your time to review the paper. The authors have revised the manuscript considering the comments provided by the reviewers and the editor. Below are the replies to the comments.

Comments from Reviewer #1

  1. The English of the paper should be improved. There are some places where grammar should be corrected. Please carefully check and correct throughout the manuscript.

Reply:

The authors thank the reviewer for this comment. The revision has been made and the revised manuscript has been checked by a native English speaker. And the English grammar has been revised and improved in this manuscript.

  1. The abstract lacks the quantitative presentation of main results. The authors are required to add more specific information regarding the effect of various parameters to this part.

Reply: Thank you for your comment. We have added add some specific information regarding the effect of various parameters in the abstract, as follows:

 The changes of settlement and horizontal deformation, geogrid strain and acceleration response of GRS retaining wall are compared and analyzed. The results show that in the dynamic test, the two wall damage modes are "wall facing outward tilt" and "wall facing outward curved". The maximum strain of the geogrid was 4.5% and 3.6% respectively, which did not reach the damage strain. The peak load is the largest mechanical response of all load factors, followed by the load magnitude and average value, and finally the load frequency.

  1. The title of the paper is: Dynamic response and geogrid strain analysis of GRS retaining wall, a large part of the paper to study the calculation method of geogrid strain, but there is no such part in the key words, it is suggested to add.

Reply:

Thank you for your valuable comments. We have replaced the “deformation characteristics” with “geogrid strain” in the keyword.

  1. Page 3: The headings of the subsections in section 2 are not clear enough, for example, the test material should be placed in a separate subsection to make it easier for the reader to read.

Reply:

Thank you for your comment. We have reclassified the titles of the subsections in section 3 as recommended.

  1. In subsection 2.1, only the grading parameters are given for the retaining wall filler, and the basic mechanical property index of sand filler is missing, and whether the sand is dried or the specific moisture content is not clearly stated when filling, please add.

Reply:

Thank you for your comment. We have added the basic mechanical parameters of soils and the water content of soils under natural conditions. This is described in section 2.2.

  1. In subsection 2.2, the dynamic load loading method (Figure 5), the meaning of Pmax is not explained, please add, in addition, it is recommended to add the waveform of half sine wave.

Reply:

Thank you for your valuable comments. We include the figure-in-diagram for illustration in Figure 3 of subsection 2.3. In the small figure, it can be seen that the test dynamic load uses a half-sine wave and marks the points where the peak load and average load values are taken in each level of loading. In the large figure, it can be seen that the dynamic load amplitude and the average value of the load alternately increase.

  1. In subsection 3.2, it is mentioned that the horizontal distance from the panel at 15cm, 75cm and 135cm cannot be found in the text with its corresponding location, so it is recommended to use the sensor's mark directly.

Reply:

Thank you for your comment. We have made the corrections as recommended. This is described in section 3.2.

  1. In Subsection 4.2, how does Figure 15 satisfy the equilibrium relation for the y-axis?

Reply:

In calculus, in order to facilitate the calculation, we can let x = x + dx and then calculate dx by other conditions. dx may be greater than 0, less than 0, or equal to 0. In Figure 15 of this paper, this is the method used.

Reviewer 2 Report

Please refer to the attached file for the Reviewer's comments and the manuscript with annotated comments.

Author Response

Comments from Reviewer #2

  1. For a large-scale test set-up, one challenge for the researcher is the preparation of the test soil with uniform density. Discuss the method of preparation of the test soil to ensure that the desired density is achieved. Discuss also the test method used to verify the density of the test soil if it meets the desired density before conducting the load test.

Reply:

The authors thank the reviewers for this earnest suggestion. Considering the length and coherence of the paper, only the quality and height of each layer of filling is controlled. The detailed process is as follows:

  1. According to the compaction formula, calculate the density of the specimen.

Where, K is the degree of compaction,  is the density of the specimen,  is the maximum dry density of the specimen.

  1. Based on the density and height of each layer, calculate the mass of each layer of soil.

3.Using the way in Figure 1, first manual tamping two times then plate compactor tamp 12-16 minutes. After reaching the specified height, use the leveling tape to level, and so on.

(a)manual tamping

(b)plate compactor tamping

Figure 1. Compacting process

  1. Explain how you would apply the results in an actual situation, for example, if the height of the retaining wall is more than the height of the experimental test box used. Is there an applicable scaling law? Is the developed equation to measure geogrid strain still applicable? Cite the limitations of the developed equation.

Reply:

The authors thank the reviewers for this earnest suggestion. The formulae in this paper are based on the theoretical formulae, which developed by elasto-plastic mechanics. It is independent of the scale of the model. In addition, the applicability and limitations of the formula have been explained in subsection 4.3 of the text.

  1. As part of the conclusion, discuss the implications of the findings for future research and potential applications.

Reply:

Thank you for your comment. The application and impact of the research results have been added as recommended. It can be seen in the conclusion:

“This paper is based on a large-scale model test to investigate the dynamic characteristics of GRS retaining walls. The results of the study contribute to reveal the effects of peak, average, amplitude and frequency of loads on the mechanical response and damage characteristics of GRS retaining walls under dynamic loading. The main conclusions are summarized as follows: ”

  1. The acronym GRS in the title should be written out. Within the manuscript, write out the first in-text reference to an acronym, followed by the acronym itself written in capital letters and enclosed by parentheses.

Reply:

The authors thank the reviewer for this comment. GRS(Line 14 ), DSC(Line 76 ) and MTS(Line 152 ) has been changed as recommended in the manuscript.

  1. Some sentences are very long. Consider breaking the long sentence into several sentences so that it is easy to read and the flow of thought can easily be understood.

Reply:

The authors thank the reviewer for the kind recommendations. Long sentences(Line 200,Line 278, Line 288 ) has been changed as recommended in the manuscript.

  1. Provide pictures of the test set-up after loading tests to support the discussion of section. 3.1 Lateral deformation of wall facing and the data presented in Figure 7.

Reply:

The authors thank the reviewer for this comment. We are very sorry that we only took photos of the top of the wall at the end of the test, as shown in Figure 2.

Due to the lateral displacement of the panel of less than 2cm maximum, on the whole, these changes cannot be observed by eye. Therefore, it is not very helpful to analyze the lateral deformation of wall facing.

(a) Equal amplitude loading

(b) Equal frequency loading

Figure 2. The end of the experiment
